# The Effect of Manual Therapy on Psychological Factors and Quality of Life in Lumbar Disc Herniation Patients: A Single Blinded Randomized Clinical Trial

**DOI:** 10.3390/ijerph21091234

**Published:** 2024-09-18

**Authors:** Burhan Taşkaya, İsmail Taşkent, Mahmut Çakıllı, Öznur Yılmaz

**Affiliations:** 1Vocational School of Health Services, Muş Alparslan University, Muş 49250, Turkey; 2Department of Radiology, Kastamonu University, Kastamonu 37150, Turkey; dr.taskent@gmail.com; 3Department of Physical Therapy and Rehabilitation, Bossan Hospital, Gaziantep 27580, Turkey; mahmut.1053@gmail.com; 4Department of Physiotherapy and Rehabilitation, Faculty of Physical Therapy and Rehabilitation, Hacettepe University, Ankara 06100, Turkey; oznurtunca@yahoo.com

**Keywords:** lumbar disc herniation, manual therapy, pain, kinesiophobia, anxiety and depression, quality of life

## Abstract

**Background:** This study aimed to investigate the effect of manual therapy on pain, kinesiophobia, pain catastrophizing, anxiety, depression, and quality of life in patients with lumbar disc herniation (LDH). **Methods:** The study included 32 LDH patients. Patients were divided into the Manual therapy group (MTG—age 39.81 ± 9.45 years) and the Exercise group (EG—age 38.31 ± 9.21 years) by sealed envelope randomization. Patients were evaluated pre-study, post-study, and after a 3-month period using the McGill–Melzack Pain Questionnaire (MMPQ), Hospital Anxiety and Depression Scale (HADS), Tampa Kinesiophobia Scale (TKS), Pain Catastrophizing Scale (PCS) and Nottingham Health Profile (NSP). The exercise group received a total of ten sessions of stabilization exercises and sham spinal mobilization in five weeks, two sessions per week. In addition to the stabilization exercises, mobilization applications including Anterior-Posterior Lumbar Spinal Mobilization, Lumbar Spinal Rotational Mobilization, and Joint Mobilization in Lumbar Flexion Position, were applied to the manual therapy group. **Results:** It was found that the HADS and TKS values decreased in the MTG group compared to the pre-treatment period (*p* < 0.05), while there was no difference between these values in the EG group (*p* > 0.05). There was a statistically significant difference in the MMPQ, PCS, and NHP values after treatment in both the MTG and EG groups (*p* < 0.05). **Conclusions:** It was found that manual therapy had positive effects on psychological factors such as pain, kinesiophobia, pain catastrophizing, anxiety, depression, and quality of life in patients with LDH. **Trial registration:** NCT05804357 (27 March 2023) (retrospectively registered).

## 1. Background

Lumbar disc herniation (LDH) is a significant condition that impacts individuals’ health, work capacity, and overall quality of life. LDH has a high incidence and affects a wide range of human life [1]. Moreover, low back pain is associated with psychological disorders such as depression and anxiety, which in turn negatively affects treatment outcomes [2]. Psychiatric disorders, especially depression, are more common among patients experiencing chronic pain. Patients with chronic pain are three times more likely to develop depression compared to those without pain, and an increase in pain severity further elevates the risk of depression onset [3]. This finding demonstrates a reciprocal association between depression and chronic pain, indicating that depression and pain worsen one another and engage in comparable pathophysiological processes [4].

A high prevalence of kinesiophobia has been observed in patients with chronic low back pain. Kinesiophobia is thought to play a negative role in rehabilitation outcomes [5]. From a biopsychosocial perspective, pain and functional disability are mutually influencing factors. These factors include physiological, psychological, and social elements, and the multidimensional and dynamic interaction between them may result in chronic and complex pain. Psychosocial factors play a prominent and significant role in determining the progression of chronic pain disorders and the processes by which pain is perceived. Some studies propose that elevated levels of psychological distress and impaired cognitive function are frequently linked to diminished effectiveness of various pain–relieving treatment methods [6].

There are several treatment options available for patients with LDH. These are basically divided into two categories: surgical and conservative treatment. Conservative treatment of LDH aims to prevent the disease from progressing to the stage of interventional and/or surgical treatment and to improve complaints that negatively affect the quality of life. Conservative treatment includes patient counseling, bed rest, medication, exercise, thermotherapy, electrotherapy, traction, orthotics, lumbar school, and manual therapy [7]. Manual therapy is used to reduce pain, provide joint and tissue mobility, inhibit sympathetic reflex activity, normalize muscle tone, and dissolve adhesions. Manual therapy is the manual treatment of the spine with two different applications: manipulation and mobilization. Manipulation is a strong thrust applied passively to the joint at high velocity but low amplitude that goes beyond the physiological range of motion limit but does not exceed the anatomical limit, while mobilization is a passive maneuver that is performed more lightly to increase joint mobility and does not exceed the physiological joint limit [8].

There are studies examining the relationship between low back pain, anxiety, and depression in patients with LDH [4,9,10] and the relationship between the quality of life and LDH [11]. Many studies have been conducted on the clinical application of manual therapy methods in the treatment of LDH. Most of these studies evaluate the effects of manual therapy on pain and functional levels [12,13,14]. However, there are some studies showing that manual therapy improves the quality of life of LDH patients [15,16]. Also, in a different study, it was determined that manual therapy applied in LDH patients caused a decrease in the depression levels of patients [17]. However, studies investigating the effect of manual therapy on the quality of life and psychological factors in LDH patients are quite limited.

To emphasize that the impact of LDH on the physical condition of the patient is not limited to this. Low back pain from LDH also has a negative impact on psychological and social factors. These factors have developed interdisciplinary working strategies to minimize functional repercussions. Rehabilitation in low back pain assumes a complex treatment aimed at creating a functional individual and improving all the components of the biopsychosocial model. The aim of our study was to investigate the effects of manual therapy on the psychological factors and quality of life in LDH patients.

## 2. Methods

This study was designed as a prospective, single-blinded, randomized, controlled clinical trial. Ethical Approval for this study was obtained from the Muş Alparslan University Scientific Research and Publication Ethics Committee, with decision number 21 at the meeting held on 29 December 2020. Informed consent was obtained from all patients included in this study before it was conducted.

### 2.1. Participants

This study was conducted between January 2021 and July 2022 on patients who were diagnosed with Lumbar Disc Herniation after physical and radiological examination at the Physical Therapy Outpatient Clinic of the Muş State Hospital. The inclusion criteria of the participants were as follows: being diagnosed with LDH with Magnetic Resonance Imaging by a physical therapy physician, a pain score of at least three on the Visual Analogue Scale (VAS), persistent pain for at least eight weeks, not receiving any physical therapy application in the last six months and being between the ages of 18–65. The exclusion criteria included a history of spinal surgery, a history of autoimmune diseases (such as ankylosing spondylitis, rheumatoid arthritis, or others), spondylolysis and spondylolisthesis, spinal fractures, cardiac pathologies, a history of stroke, the presence of cauda equina syndrome, continuous use of pain medication, the presence of spinal inflammation, the presence of spinal tumors, COVID-19, and pregnancy.

### 2.2. Procedure

The sample size of the study was formed by taking five patients of each group and considering the pre- and post-treatment change levels of the McGill–Melzack Pain Questionnaire obtained from the pilot study conducted with ten patients in total. According to the two-way sample size analysis performed with the G power 3.1.9.2 program (effect size: 1.19), it was determined that 16 patients should be included in both groups at 80% power and 5% Type 1 error level.

The snowball sampling method was used as the chosen method. Patients were divided into two groups: manual therapy group (MTG) and exercise group (EG). EG patients underwent stabilization exercises and sham spinal mobilization, and MTG patients underwent stabilization exercises and spinal mobilization. The study started with 40 patients who met the inclusion criteria; the percentage of possible loss to follow-up was set at 20%. The study concluded with 32 individuals, consisting of 26 males and six females after eight people withdrew throughout the duration of the clinical study (Figure 1). Patients did not use any painkillers during the treatment.

### 2.3. Randomization and Blinding

In the trial, patients were allocated into two groups in a 1:1 ratio by the sealed envelope technique. The patients remained blinded to their group assignments. A single physiotherapist conducted both manual therapy interventions and tests throughout the study period.

### 2.4. Measurements

Demographic information was obtained, and pain, psychological factors, and quality of life were assessed. The patients included in the study were evaluated three times: pre-treatment (T1), post-treatment (T2), and a 3-month follow-up evaluation (T3).

### 2.5. Demographic Information

Pre-treatment evaluation of all patients involved in the trial included the recording of diagnosis, gender, age, height, weight, and body mass index.

### 2.6. Pain Assessment

*McGill–Melzack Pain Questionnaire (MMPQ):* The questionnaire comprises four sections. In the introductory part, the form collects the patient’s name, surname, age, medical diagnosis/problem, type and dosage of analgesics (if used), and additional information to ascertain the location, characteristic, temporal relationship, and intensity of the pain. Section One: In this section, patients are instructed to indicate the location of their pain on a body diagram and to use the letter “D” to denote deep pain, “S” for superficial pain, and “DS” for pain that is both deep and superficial. Section Two: This section contains 20 sets of words exploring pain in sensory, perceptual, and evaluative aspects. Each set consists of 2–6 words describing different facets of pain. Patients are asked to choose the word cluster that corresponds to their pain and to mark the word that best describes their pain within the chosen cluster. Section Three: This section examines the temporal relationship of pain, featuring word groups to determine pain continuity, frequency, and factors that may increase or decrease pain. Section Four: In this part, there are five-word groups to assess pain intensity, ranging from “mild” to “unbearable” pain, and six questions to determine the tolerable pain level—also referred to as “livable” or “target pain”—that the patient can experience without discomfort. In summary, the McGill–Melzack Pain Questionnaire measures the location, temporal relationship, intensity, sensation, and tolerable pain level experienced by the patient [18].

### 2.7. Psychological Factors Assessment

*Hospital Anxiety and Depression Scale (HADS):* This self-assessment scale is employed to identify the risk of anxiety and depression in patients and to measure the level and severity change. The scale comprises 14 questions in total, with seven dedicated to measuring anxiety and the remaining seven assessing depression. The lowest score that patients can get from both subscales is 0, and the highest score is 21 [19]. A Turkish validity and reliability study of the Hospital Anxiety and Depression Scale was conducted by Aydemir Ö. et al. [20].

*Tampa Kinesiophobia Scale (TKS):* This 17-question scale evaluates individuals’ fear of re-injury during movement. The scale utilizes a 4-point Likert scoring system, with patients receiving a minimum score of 17 and a maximum score of 68 points. A higher score indicates a greater degree of kinesiophobia. In research, scores of 37 points or above are considered indicative of high kinesiophobia [21]. A Turkish validity and reliability study of the Tampa Kinesiophobia Scale was conducted by Yılmaz Ö. T. et al. [22].

*Pain Catastrophizing Scale (PCS):* This scale reliably measures specific variables such as fears, emotions, or thoughts associated with individuals’ past pain experiences, intense pain, disability, and emotional disturbances. The 13-question scale is scored from 0–4 (0 = Never, 1 = Somewhat, 2 = Moderately, 3 = Seriously, 4 = Always). The overall score range is between 0–52. An increased score on the scale signifies a heightened fear of experiencing pain [23]. A Turkish validity and reliability study of the Catastrophic Pain Scale was conducted by Uğurlu M. et al. [24].

### 2.8. Assessment of the Quality of Life

*Nottingham Health Profile (NHP):* This valid and reliable quality-of-life scale is utilized to evaluate the physical, emotional, and social effects of diseases on individuals. It consists of six sections—pain, physical activity, energy, sleep, social isolation, and emotional reactions—and a total of 38 questions. The score of each statement in each sub-dimension is different. When scoring, the statements belonging to that dimension are summed under each sub-dimension, and the scoring of each sub-dimension is between “0” and “100” [25].

### 2.9. Intervention

For EG patients, stabilization exercises and sham spinal mobilization were administered, while MTG patients received both stabilization exercises and spinal mobilization procedures. The treatment was implemented twice a week for five weeks, totaling ten sessions. Upon the completion of the treatment, stabilization exercises were recommended as a home exercise program until the follow-up evaluation in the third month post-treatment. Weekly phone calls were made to monitor the home program.

*Stabilization exercises:* These workouts aim to reactivate and fortify the muscles responsible for long-term spinal column stabilization. The focus is also on preserving the neutral zone’s stabilization capacity for the lumbar spine’s tonic muscle control. Transversus abdominis and multifidus muscles are engaged as deep core muscles in this approach. The stabilization exercise program was carried out in three phases, with progression based on patient improvements. Phase 1 exercises: The primary objective of this phase is to enhance the simultaneous contraction ability of deep stabilizer muscles independent of global muscles. Exercises in this phase consist of local muscle activation while maintaining a supine neutral position, local muscle activation while maintaining a prone neutral position, piriformis stretching in a supine position, supine trunk rotation to the left and right, and supine bridging. Phase 2 exercises: Extremity movements were incorporated into the stabilization program to reinforce local stabilizer muscle activation and coordination with global stabilizer muscles. Exercises in this phase included supine hip-knee flexion, supine single-leg bridging, prone diagonal arm-leg raises, and diagonal arm-leg raises in a crawling position [26]. Phase 3 exercises: The goal of this phase was to enhance balance and coordination, integrating local muscle activation into functional movements. As a result, less stable surfaces were chosen for the exercises. Exercises in this phase included standing mini squats, leg extension on a ball, side bridge on a ball, and knee extension with an exercise band in a crawling position [27].

Manual therapy practices:

Standard Maitland grade IV mobilization was applied. We applied all our maneuvers at the pain limit.

*Anterior-Posterior Lumbar Spinal Mobilization:* The patient lies prone, and the spinal processes are palpated. The ulnar edge of the hand is positioned over the spinal process, and the thumb web space of the other hand is placed over the fingers to strengthen the hand. The elbows are kept straight, and downward pressure is applied. Low-speed, physiologically limited repetitive pushing is applied [28]. This was applied to each lumbar vertebra with 20 repetitions.

*Lumbar Spinal Rotational Mobilization:* The patient lies on their side with the painful side facing up. The upper hip and knee are flexed at 90 degrees to aid in rotational stress, and the lower leg is extended. The shoulder is firmly pulled so that the pelvis is positioned forward while the shoulder is positioned backward. Stand in front of the patient. One hand stabilizes the shoulder, while the hard part of the other palm is placed on the ilium wing, forearm horizontal, and fingers facing you. With the hand on the ilium, a rotational force is applied by exerting pressure in a horizontal direction toward you. Slow and repetitive movements are performed in this position [29]. Slow and 20 repetitions were performed in this position. The same application was performed for both sides.

*Joint Mobilization in the Lumbar Flexion Position:* In the prone lumbar flexion position on the stretcher, mobilization was applied to the entire lumbar region with one hand on the sacrum and the other hand at the end of the thoracic vertebrae. Then, the vertebrae were mobilized one by one. The caudal hand pushed the lower vertebra towards the transverse plane, while the cranial hand pushed the upper vertebra superiorly. The application was performed on each lumbar vertebra with 20 repetitions.

*Sham Spinal Mobilization:* In the spinal mobilization applications mentioned above, only hand contact is made, but no force is applied.

### 2.10. Statistical Analysis

The analysis of the data was conducted using the SPSS 25 (Statistical Package for Social Sciences) software for Windows. To assess the adherence of the data to a normal distribution, skewness, and kurtosis values were examined. Skewness–kurtosis values ranging between (−1.5) and (+1.5) fulfill the assumptions of normality [30]. Upon conducting the normality analysis, it was found that all data conformed to a normal distribution. Chi-square tests were employed to assess the homogeneity of the demographic variables in the MTG and EG groups, while independent samples *t*-tests were used to assess continuous variables. For the assessment of MMPQ, HADS, TKS, PCS, and NHP, results were compared between the MTG and EG groups using an independent sample *t*-test. Within-group comparisons of means (T1, T2, and T3) were performed using repeated-measures analysis of variance (ANOVA). In cases where significant differences were found in the within-group comparisons, the Least Significant Difference (LSD) test was used to identify the source of the variance. Data were examined with a 95% confidence interval, considering a *p*-value of <0.05 as statistically significant.

## 3. Results

Demographic characteristics such as age, weight, height, and BMI were found to be similar in the patients who participated in the study. The distribution of the demographic information of the patients according to the groups is presented in Table 1.

In both MTG and EG patients included in the study, 81.2% were male, and 18.8% were female. It was found that the demographic characteristics of the individuals in the MTG and EG groups were homogeneously distributed, and there was no difference between the groups (*p* > 0.05) (Table 1).

Intragroup and intergroup comparisons of the McGill–Melzack Pain Questionnaire, Hospital Anxiety and Depression Scale, Tampa Kinesiophobia Scale, Pain Catastrophizing Scale, and Nottingham Health Profile are shown in Table 2.

There was no difference between the groups in the HADS and TQS values before treatment (*p* > 0.05). It was found that only the HADS and TKS scores of the patients in the MTG decreased after the application, and this created a significant difference (*p* < 0.05).

It was found that the PCS score was higher in the MTG than the EG before the treatment (*p* < 0.05), the scale scores decreased in both groups after the treatment, and this constituted a significant difference (*p* < 0.05). However, it was observed that the effect size was more than twice that of the EG in the MTG.

A significant difference was found in the intra-group scores of the MMPQ and NHP values of the patients in both groups (*p* < 0.05). Post hoc analysis for both groups showed that the difference was due to the T1 score.

## 4. Discussion

This study is the first single-blinded randomized clinical trial to examine the effect of manual therapy on the psychological factors and quality of life in LDH patients. In our study, significant improvements were seen in kinesiophobia, anxiety, and depression in the group treated with manual therapy. Significant improvements were found in pain, quality of life, and pain catastrophizing in both groups. These results revealed the positive effect of manual therapy on pain, psychological factors, and quality of life.

With the manual therapy we applied, a significant decrease in the pain levels of patients were observed. In manual therapy, mechanical stimuli are generated through manual contact and pushing on the joints. This activates the gate control mechanism and inhibits the impulse transmission of the thin A-delta and C fibers that carry pain stimuli by activating the thick myelinated fibers (A-β) that activate the mechanoreceptors in the joints and are responsible for carrying these sensations [31]. Furthermore, our manual therapy approach for the lumbar spine produces afferent stimuli by stretching the joint capsule to alleviate spinal facilitation resulting from the hypersensitivity of a damaged segment. Manual therapy induces stretching of the joint capsule, which activates sensory signals to the spinal cord via proprioceptors within the capsule and muscles that stimulate the periaqueductal grey matter (PAG) of the midbrain. This activation of the descending serotoninergic and noradrenergic system results in analgesia [32,33]. The activation of these two mechanisms is thought to help reduce pain levels in patients. Similar to our study, some studies in the literature report that manual therapy and exercise applied to LDH patients resulted in a significant reduction in pain [34]. Again, a different study reported that patients who underwent manual therapy had significant reductions in low back pain in a short period [35].

Prior research has established that the psychosocial condition has a prominent and unambiguous function in the emergence of pain [36]. Depression and fear have been proven to be associated with perceived improvement, pain, and function in patients with low back pain [37]. Martí-Salvador M et al. found significant improvements in the HADS, PCS, and Fear-Avoidance Beliefs Questionnaire scores of patients with low back pain after three months of manual therapy [38]. Again, in a different randomized clinical study, ten sessions of manual therapy were applied to patients, and significant differences were observed in terms of depression and kinesiophobia in the manual treatment group [39]. Manual therapy applied in patients with chronic low back pain was found to cause significant changes in the Fear-Avoidance Beliefs Questionnaire [40]. The results of the current study are similar to those of the literature. However, unlike the results of our study, Sung et al. found that manual therapy and exercise applied to patients with low back pain did not cause a significant difference in the Fear-Avoidance Beliefs Questionnaire. It is thought that this difference may be due to two reasons: first, manual therapy was applied to the thoracic vertebrae in the mentioned study, while manual therapy was applied to the lumbar vertebrae in our study. Second, while manual therapy was performed in a single plane in the mentioned study, manual therapy was performed in three different planes in our study. These results show that the location of the manual therapy site and the manual therapy technique can also affect the treatment results.

Catastrophizing pain is defined as a negative cognitive-emotional response to a current or future painful situation that tends to exaggerate and magnify the pain sensation [41]. A study in which three different treatments were applied to patients with chronic neck pain found that there was a significant decrease in the PCS value in the manual therapy group [42]. In a different study, it was stated that mobilization and manipulation as a method of manual therapy applied to fibromyalgia patients caused significant changes in PCS values, and this condition was maintained. A systematic review examined the evidence for the effectiveness of manual therapy in catastrophizing, fear-avoidance, kinesiophobia, and pain in individuals with chronic musculoskeletal disorders. This review found that manual therapy was effective in fear avoidance, kinesiophobia, and catastrophic pain [43]. It is thought that the positive effect of the manual therapy method we applied in our study on pain perception helps to have positive effects on kinesiophobia, pain catastrophizing, anxiety, and depression, which are related to pain.

Health-related quality of life was found to decrease with the occurrence of low back pain, leg pain, or both after LDH [44]. Positive depression results have been found to be associated with poor quality of life in LDH patients [45]. In a study conducted on LDH patients, patients were divided into two groups: one group received exercise and manipulation, and the other group received exercise and mobilization. Similar to our study, significant changes in the quality of life occurred in the mobilization group [46]. In a study conducted on patients with chronic low back pain, stabilization exercises were applied to one group, and manual therapy methods such as soft tissue mobilizations, muscle energy techniques, joint mobilization, and/or manipulations to the other group; significant improvements in the quality of life were found in both groups, and no difference was found between groups [47]. The results of our study show that manual therapy has positive effects on pain and psychological factors as well as contributing to improving the quality of life.

Manual therapy is generally applied to eliminate the pathological barrier that causes functional movement in peripheral or spinal joints. It is used in the treatment of many symptoms and diseases in the clinic. In a systematic review of randomized controlled trials in neck pain, mobilization and manipulation, which are manual therapy methods combined with exercise, were found to be effective [48]. In another systematic review, manual therapy applications such as manipulation, mobilization, and massage were recommended in patients with lumbar spinal stenosis [49]. In addition to many methods, manipulation is used in the treatment of tension headaches, cervicogenic headaches, and migraine, which are the most common types of headaches [50]. In addition, the manual therapy method, which includes mobilization and manipulation, is also useful in osteoarthritic conditions with hip pain, joint motion limitation, and capsular end sensation [51].

The biggest limitation of this study is that the treatment period of the patients coincided with the COVID-19 pandemic, which caused us to have difficulty in finding participants. However, selecting patients from the university staff and our close circle, as well as treating some patients at home, helped us overcome these limitations. The most powerful aspect of our study is that it is the first study in our knowledge to examine the effect of manual therapy applications on the psychological factors and quality of life in LDH patients. We believe that the results of our study will be a guide for future studies using different treatment applications and manual therapy in LDH.

## 5. Conclusions

This study found that manual therapy had positive effects on pain, depression, anxiety, kinesiophobia, pain catastrophizing, and quality of life in patients with LDH.

## Figures and Tables

**Figure 1 ijerph-21-01234-f001:**
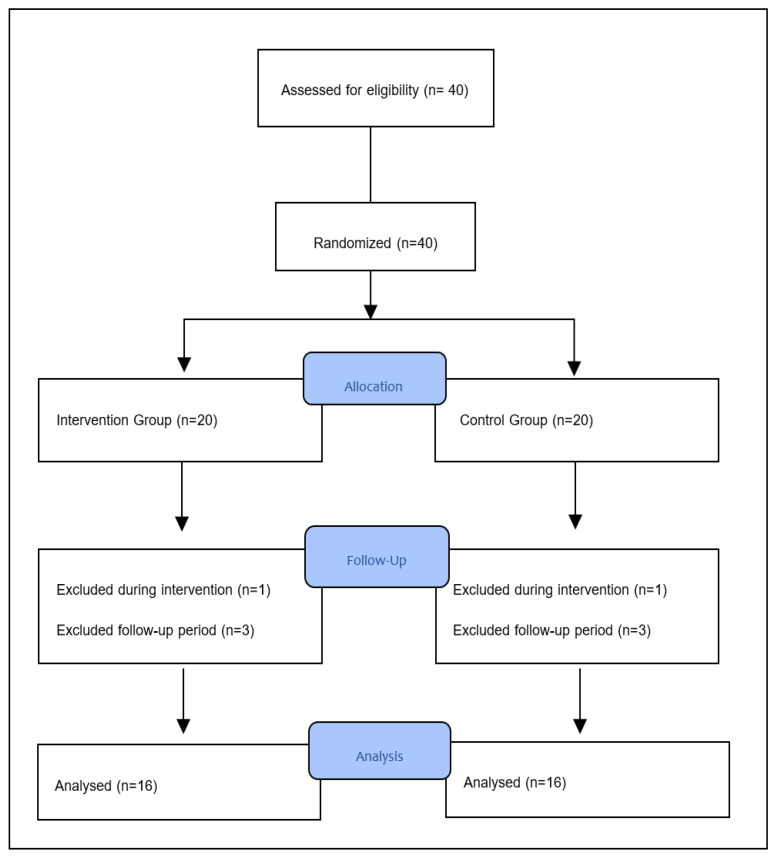
Flow diagram.

**Table 1 ijerph-21-01234-t001:** Distribution of the demographic information by groups.

Continuous Variables	MTG (n = 16) (X ± SD)	EG (n = 16) (X ± SD)	t	*p* *
Age (years)	39.81 ± 9.45	38.31 ± 9.21	0.455	0.653
Weight (kg)	89.62 ± 12.96	80.75 ± 11.55	2.044	0.050
Height (cm)	175.94 ± 9.45	170.50 ± 7.00	1.848	0.074
BMI (kg/m^2^)	28.96 ± 4.27	27.69 ± 2.82	0.989	0.330

* Independent sample *t*-test. MTG (Manual therapy group), EG (Exercise group), BMI (Body Mass Index).

**Table 2 ijerph-21-01234-t002:** Intragroup and intergroup comparison of the psychological factors and quality of life assessment.

Variables	MTG (16)	EG (16)	Test Ve *p*-Value *
**McGill–Melzack Pain**			
**Questionnaire**	58.56 ± 11.17	40.62 ± 9.85	t: 4.814 *p*: 0.000
**T1**	28.62 ± 10.72	23.43 ± 11.51	t: 1.318 *p*: 0.197
**T2**	32 ± 15.69	21.31 ± 11.71	**t: 2.182 *p*: 0.037**
**T3**	Mauchly’s W: 0.735 ^+^	Mauchly’s W: 0.686^+^	
**Test and *p*-value**	**F: 45.816 *p*: 0.000**	**F: 21.459 *p*: 0.000**	
**Post hoc**	T1 > T2,T3	T1 > T2,T3	
**Effect size**	0.753	0.589	
**Hospital Anxiety and Depression**			
**Scale**	12.93 ± 6.22	10.68 ± 6.74	t: 0.981 *p*: 0.334
**T1**	6.62 ± 5	9.43 ± 5.76	t: −1.474 *p*: 0.151
**T2**	9.06 ± 5.35	9.18 ± 6.03	t: −0.062 *p*: 0.951
**T3**	Mauchly’s W: 0.523 ^++^	Mauchly’s W: 0.967 ^+^	
**Test and *p*-value ****	**F: 10.930 *p*: 0.000**	F: 1.966 *p*: 0.154	
**Post hoc**	T1,T3 > T2	-	
**Effect size**	0.422	0.117	
**Tampa Kinesiophobia Scale**			
**T1**	40.62 ± 7.67	37.68 ± 7.21	t: 1.115 *p*: 0.274
**T2**	35.43 ± 6.41	31.81 ± 9.36	t: 1.278 *p*: 0.211
**T3**	35.93 ± 7.11	34.56 ± 6.02	t: 0.590 *p*: 0.560
**Test and *p*-value ****	Mauchly’s W: 0.714 ^+^	Mauchly’s W: 0.658 ^+^	
	**F: 4.951 *p*: 0.014**	F: 2.510 *p*: 0.098	
**Post hoc**	T1 > T2,T3	-	
**Effect size**	0.248	0.143	
**Pain Catastrophizing Scale**			
**T1**	26.12 ± 10.84	14.43 ± 12.62	**t: 2.809 *p*: 0.009**
**T2**	12.93 ± 10.06	8.18 ± 9.68	t: 1.360 *p*: 0.184
**T3**	10.43 ± 8.45	8.18 ± 12.01	t: 0.613 *p*: 0.545
**Test and *p* value**	Mauchly’s W: 0.653 ^++^	Mauchly’s W: 0.854 ^+^	
	**F: 20.836 *p*: 0.000**	**F: 5.444 *p*: 0.013**	
**Post hoc**	T1 > T2,T3	T1 > T2,T3	
**Effect size**	0.581	0.266	
**Nottingham Health Profile**			
**T1**	187.13 ± 116.55	143.27 ± 108.71	t: 1.101 *p*: 0.280
**T2**	67.79 ± 64.65	92.22 ± 83.66	t: −0.924 *p*: 0.363
**T3**	83.44 ± 89.41	65.13 ± 76.18	t: 0.623 *p*: 0.538
**Test and *p*-value ****	Mauchly’s W: 0.605 ^++^	Mauchly’s W: 0.778 ^+^	
	**F: 10.096 *p*: 0.002**	**F: 10.602 *p*: 0.000**	
**Post hoc**	T1 > T2,T3	T1 > T2,T3	
**Effect size**	0.402	0.414	

* Independent sample *t*-test ** Repeated Measure ANOVA test ^+^ “F” values in this table are Greenhouse–Geisser F values. ^++^ The “F” values in this table are Sphericity Assumed F values. MTG (Manual therapy group), EG (Exercise group), T1 (pre-treatment), T2 (post-treatment), T3 (3-month follow-up evaluation).

## Data Availability

Data are not publicly available but can be obtained from the corresponding author on request.

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
