# Peer review of "The Effect of Manual Therapy on Psychological Factors and Quality of Life in Lumbar Disc Herniation Patients: A Single Blinded Randomized Clinical Trial"

_ijerph, 2024, doi:10.3390/ijerph21091234_

Round 1

Reviewer 1 Report

Comments and Suggestions for Authors

Comments and Suggestions for Authors

Thank you for the opportunity to examine this interesting work that investigates the effects of manual therapy on psychological factors and quality of life in patients with lumbar disc herniation.-I would like to kindly remind you to carefully review your article according to the iThenticate report. The duplication rate is 30%.

-Complete the affiliation addresses and standardize the name format.

INTRODUCTION

-“mobilization is a passive maneuver that is performed more lightly to increase joint mobility and does not exceed the physiological joint limit”. Add that Manipulation also does not exceed the physiological joint limit.

MATERIALS AND METHODS

-Develop the abbreviation MRI the first time it appears in the text.

-A figure of the flow chart of this study conducted according to Consolidated Standards of Reporting Trials (CONSORT) guidelines should be added.

-Why are “score of at least three on the Visual Analogue Scale (VAS)”, “the ages of 18-65” considered inclusion criteria? Explain it. The time of evolution of the pain (acute, subacute, chronic) should also be taken into account.

-2.2. Procedure: This section reflects the calculation of the sample size and the two treatment groups. It would be better to have a section that only talks about the calculation of the sample size and another section about the procedure with the two treatment groups.

-For the calculation of the sample size, was the % of possible loss to follow-up taken into account? Indicate it.

-How was it controlled that the patients did not take analgesia during the study?

-2.3. Randomization and blinding: Clearly state who is blinded. Even if the patient does not know the group to which he has been assigned, one of the groups receives stabilization exercises in addition to spinal mobilization. When the patient signs the informed consent, he knows that there are two different treatment groups. I find it difficult to blind the patient.

-2.6. Pain Assessment McGill-Melzack Pain Questionnaire: when the patient describes his pain, how is it determined that its origin is in the disc?

-2.7. Psychological Factors Assessment: Indicate how this scale scores.

-Indicate whether all scales used are adapted and validated in the Turkish population. Also, indicate whether they are sensitive and reliable tests.

-2.8. Assessment of Quality of Life: Indicate how this scale scores.

-2.9. Intervention: Clarify whether the 3rd assessment was the 3rd month after finishing the treatment, or from the beginning of the sessions.

-Anterior-Posterior Lumbal Spinal Mobilization and other maneuvers: these were applied to each lumbar vertebra with 20 repetitions. Was pain behavior taken into account during the maneuvers? How?

RESULTS

-TABLE 2: review “Test ve p değeri*”.

-Explain all abbreviations in the Table legend.

DISCUSSION

-When manual therapy is mentioned in lines 311-318, it should be specified which manual therapy procedures these studies refer to.

-Indicate in this section both the strengths and limitations of this study that could affect the results of the same. As well as possible lines of future research.

CONCLUSIONS

They should respond only to the objectives set.

REFERENCES

-It is recommended to review the format of the references and adapt it to the standards of the journal.

-30% of the references are obsolete. It is recommended that they be replaced by more current ones.

-Journal titles should be abbreviated.

Reviewer 2 Report

Comments and Suggestions for Authors

Dear editor of the journal

Thank you for choosing me as a reviewer

This article tries to investigate the effect of exercises on psychological aspects 

An important point that the authors must answer in the meantime is why physical factors have not been worked on for patients with disc problems.

In the abstract, it is better to mention the intensity of the exercises. The definitive results should be mentioned in the results section and a better conclusion should be mentioned.

The importance and necessity of the work is not well mentioned in the introduction

Mention the sampling method, is the sample available

The validity and reliability of the questionnaires should be mentioned in full

In the discussion section, the reasons for recovery and effectiveness or complete lack of effectiveness should be mentioned.

Reviewer 3 Report

Comments and Suggestions for Authors

The paper seems promising but might benefit from some revisions, particularly in the clarity of the methodology and the robustness of the results and discussion sections. At this stage, I would recommend minor revisions before acceptance.   I have other comments:

- In Table 01, I need help understanding the formula on the right.

- Please avoid using abbreviations in the abstract.

- in Table 2, on the right, there is a word in the Turkish language!

Round 2

Reviewer 1 Report

Comments and Suggestions for Authors

The paper has improved considerably in quality, however, some aspects need further improvement.

Comments 2: A figure of the flow chart of this study conducted according to Consolidated Standards of Reporting Trials (CONSORT) guidelines should be added.

Response 2: Thank you for this important contribution. At the end of page 3, the following figure has been added.

Comments: just to comment that “Figure 1. Flow diagram” should appear as a caption.

Comments 7: 2.3. Randomization and blinding: Clearly state who is blinded. Even if the patient does not know the group to which he has been assigned, one of the groups receives stabilization exercises in addition to spinal mobilization. When the patient signs the informed consent, he knows that there are two different treatment groups. I find it difficult to blind the patient

Response 7: Thank you for this very valuable contribution. We told the patients we included in the study that we would apply different treatments to LDH patients, and information about the treatment they would receive was written in the informed consent of each group. Because of this situation, the patients in the two groups were blinded to each other.

Comments: Indeed, if the patient has been properly informed about the two different treatments, blinding is not possible because they can perfectly know which treatment they are receiving. Blinding could occur if a manual therapy placebo was applied to the control group.

REFERENCES

Comments 1: -It is recommended to review the format of the references and adapt it to the standards of the journal.

Response 1: I have organized all references according to the standard of the journal

Comments: References should follow this format:

  • Journal Articles:
    1. Author 1, A.B.; Author 2, C.D. Title of the article. Abbreviated Journal Name YearVolume, page range.
  • Books and Book Chapters:
    2. Author 1, A.; Author 2, B. Book Title, 3rd ed.; Publisher: Publisher Location, Country, Year; pp. 154–196.
    3. Author 1, A.; Author 2, B. Title of the chapter. In Book Title, 2nd ed.; Editor 1, A., Editor 2, B., Eds.; Publisher: Publisher Location, Country, Year; Volume 3, pp. 154–196.

Reviewer 2 Report

Comments and Suggestions for Authors

File edited and its can be publish
